# Comparison of the Effect of Cold Plasma with Conventional Preservation Methods on Red Wine Quality Using Chemometrics Analysis

**DOI:** 10.3390/molecules27207048

**Published:** 2022-10-19

**Authors:** Iwona Niedźwiedź, Vasil Simeonov, Adam Waśko, Magdalena Polak-Berecka

**Affiliations:** 1Department of Microbiology, Biotechnology and Human Nutrition, University of Life Sciences in Lublin, 8 Skromna Street, 20-704 Lublin, Poland; 2Faculty of Chemistry and Pharmacy, University of Sofia, 1 James Bourchier Blvd., 1126 Sofia, Bulgaria

**Keywords:** cold plasma, phenolic compounds, antioxidant activity, chemometric analysis, wine preservation, red wine, shelf life

## Abstract

In this study, the effect of cold plasma (CP) on the physicochemical and biological properties of red wine was investigated in comparison with the effects of the conventional preservation method and the combined method. In addition, the effect of storage time after the application of each of the analyzed methods was evaluated. The study examined the effects of the different preservation methods on the pH, color, phenolic content, antioxidant activity and microbiological purity of the red wine. Chemometric analysis was used to discover the relationship between the preservation method used and wine quality. In the wine samples tested, a reduction in phenolic compounds and a decrease in antioxidant activity were noted after storage. This effect was mildest for preservation methods with the addition of potassium metabisulphite and those in which a mixture of helium and nitrogen was used as the working gas. On a positive note, the CP treatment did not affect the color of the wine in a way perceptible to the consumer: ∆E*—1.12 (He/N_2_; 5 min). In addition, the lowest growth of microorganisms was detected in the CP-treated samples. This indicates the potential of cold plasma as an alternative method to the use of potassium metabisulfite in wine production, which may contribute to its wider use in the alcohol industry in the future.

## 1. Introduction

Wine is a traditional alcoholic beverage, which has been present in the cultures of the whole world for thousands of years. The first references to winemaking can be found in records from Georgia dating back to 6000 BC. For centuries, the culture of wine production and consumption has been developing in the hot-climate areas today occupied by Greece, Italy, Spain and Portugal, which are some of the most important winemaking countries in Europe [1]. In recent years, climatic changes have contributed to increased grapevine cultivation and wine production also in the hitherto cold Western regions of the continent [2]. Large-scale production of wine all over the world has resulted in a wide range of wines being available on the market.

On the consumption end of the wine business, growing consumer awareness makes customers attach increasing importance to the quality of the wine they drink, especially its physical, chemical and organoleptic properties [3]. The main parameters consumers assess are alcohol content, composition, acidity, color, aroma and taste. In addition, more and more importance is given to the health-promoting properties of wine [4], in particular red wine, which is considered to have some potent health benefits. The salutary effects of red wine are due to its complex chemical composition, which, in addition to water, sugars and alcohol, includes bioactive compounds. Among the most important bioactive compounds are polyphenols, which are characterized by antioxidant [5] and antibacterial properties [6], and have a beneficial effect on the nervous and cardiovascular systems [7]. Many epidemiological studies indicate that moderate red wine consumption contributes to a reduction in mortality, protects individuals against diabetes [8] and has a preventive effect on certain cancers [9]. 

The final quality of wine depends on the numerous chemical reactions occurring in the complex food matrix during the many stages of the winemaking process that include fermentation, preservation and aging. The fermentation of grapes involves a large number of microorganisms, whose uncontrolled growth can lead to the deterioration of the product or even its complete spoilage. To avoid this, different preservation methods are used, which prevent the development of undesirable microflora and extend the shelf life of the product [10]. A common preservative used for this purpose is sulfur dioxide (SO_2_), which exhibits antimicrobial and antioxidant properties and prevents the non-enzymatic oxidation of wines [11]. However, this compound can have a negative impact on the quality of wine by causing sensory changes and it can also be potentially harmful to consumers’ health as it can lead to headaches, dermatitis or diarrhea [12]. For this reason, regulations have been introduced to determine the acceptable dose of SO_2_ in wine production. However, wines without the addition of sulphates are considered healthier, hence the growing interest in new preservation methods, which would minimize or eliminate the use of sulphur compounds. The use of thermal technologies in wine preservation is not appropriate, as they can adversely affect the taste, color and aroma of wine through the loss of bioactive compounds when exposed to the temperature used. With this in mind, researchers are currently particularly interested in new non-thermal methods of food preservation, such as high hydrostatic pressure (HHP) or pulsed electric field (PEF), which, in addition to providing microbiological safety, due to the low temperature of the process, have less impact on product quality, e.g., by not contributing to high losses of thermolabile compounds [13]. 

Such methods also include cold plasma, which is considered to be the fourth state of matter [14]. The concept of plasma was introduced in 1928 by the American physical chemist Irving Langmuir to describe a low-pressure electrical discharge [15]. Food technology makes use of low-temperature non-equilibrium plasma, which is characterized by a thermal imbalance between electrons and other gas molecules such as ions, inert particles or radicals. The lower temperature of the electrons relative to the heavy gas molecules allows the food preservation process to be carried out at temperatures close to ambient [16]. Plasma components, whose amount and type depend on the working gas used, enable, as indicated by numerous scientific reports, the effective elimination of microorganisms from the food matrix [2,17]. The mechanism of microbial inactivation using cold plasma is still not fully understood despite many studies [18,19]. However, it is believed that the reactive compounds, UV photons, and excited and ground state atoms produced during cold plasma generation can disintegrate microbial surface structures or degrade genetic material [20,21]. In the context of the use of cold plasma in the food industry, it is worth noting its advantages, such as the short duration of the process, low temperature, minimal water consumption and effective decontamination, which allow this method to be viewed as a sustainable food production technology [22].

Due to increasing consumer expectations of product quality and microbiological safety, it is necessary to search for alternative preservation techniques. This is particularly important in the alcoholic beverage industry, where the application of conventional methods using high temperatures is undesirable. The present work is a continuation of our previous research [2], in which we qualitatively and quantitatively determined phenolic compounds and biogenic amines in wine samples subjected to different preservation treatments. The methods studied were the conventional preservation method (addition of potassium metabisulfite at 30 mg/L or 100 mg/L), cold plasma and a combined method (cold plasma and addition of potassium metabisulfite at 30 mg/L). In the present study, the scope of the research was extended to analyze the effects of the above-mentioned preservation methods and storage time on the physicochemical properties (pH, color, total polyphenol content, total anthocyanin content, antioxidant properties) and microbiological safety of red wine.

## 2. Results and Discussion

### 2.1. Physicochemical Properties

Physicochemical properties such as pH, color, total phenolic compounds, total anthocyanin content and antioxidant properties play a crucial role in wine quality assessment. Therefore, in this study, we investigated how these parameters were affected by three different preservation processes (cold plasma, a method combining cold plasma with the addition of 30 mg/L potassium metabisulfite, and the traditional method involving the addition of 100 mg/L potassium metabisulfite) and storage time (Table 1 and Table 2).

#### 2.1.1. Determination of pH and Color Measurement

Analyzing the influence of different preservation processes on the pH value of wine samples, we did not observe major changes in acidity (Table 1). The pH was ~3.52 before storage and ~3.49 after storage. These values were similar to the pH of the control sample and did not differ from the acidity of commercially available red wines [23]. 

The effect of the different preservation methods and storage time on wine color was evaluated using the CIELab model. An analysis of color lightness (parameter L*) showed that, in non-stored samples, the value of this parameter was higher than in the control. The highest L* values were recorded in the samples treated with the combined method—15.13 (cold plasma: 5 min, He/N_2_), and the traditional method—14.94. When analyzing the a* parameter indicating the redness of the product, we observed the same dependencies. Higher values of this parameter were recorded in samples where the addition of potassium metabisulfite was used (in a sample with addition of potassium metabisulfite at a dose of 30 mg/L–45.89 and 100 mg/L–46.14). In general, samples preserved with cold plasma using helium/nitrogen as the working gas and samples with the addition of potassium metabisulfite were characterized by higher values of this parameter. The red color of wine is mainly due to anthocyanins. SO_2_ can form bonds with them, which directly contributes to the increased brightness of the product. From the point of view of commercial wine quality assessment, this is an undesirable effect [13]. 

After three months, samples 17, 19, 24, 26 and 29 were darker in color compared to the control (about 9.35; 13.44; 41.23; 20.39; and 15.94%, respectively). Cold plasma (working gas: He/O_2_) was used to preserve four of them, which may indicate the involvement of oxygen and its reactive compounds in the process of wine darkening. The mechanism of this phenomenon may be related to ozone and hydroxyl radicals resulting from cold plasma generation. These compounds can cause oxidative cleavage of chromophores, leading to the breakdown of anthocyanins and loss of product color [24]. However, it should be noted that the 5 min treatment with cold plasma alone (He/N_2_) as well as with the addition of potassium metabisulfite (30 mg/L) did not cause any change in the value of the L* parameter compared to the control sample. A slight darkening of wine under cold plasma was also observed by Lukić et al. (2019) [25], who studied the effect of cold plasma generated by high voltage pulsed power supply (working gas: argon, treatment time: 2, 5 or 10 min.) on white and red wine. Those authors observed a decrease in the value of the L* parameter with increasing time of exposure to cold plasma. Analyzing the other parameters that comprise the CIELab model (a*; b*), we found the same relationships as in the case of the brightness parameter. The highest value of the parameter indicating the redness of the product (a*) was recorded in the sample with the addition of potassium metabisulfite at a dose of 100 mg/L and was 47.60. An increase in the parameter b* indicates a greater proportion of yellow color in the wine, which may be due to the oxidation of flavanols. Conversely, an increase in C* indicates an increase in the clarity of the product, which may be due to sedimentation of colloids in the wine during storage [26]. Nevertheless, data explaining the effect of cold plasma on wine color are still lacking, and the results are inconclusive.

In order to determine whether the changes we observed could affect the assessment of wine quality by the consumer, the parameter ∆E* was calculated, which indicates whether the differences in color between the control sample and the test sample can be perceived by the human eye (∆E* ≥ 3) [27]. In samples that were tested immediately after exposure to a sterilizing agent, the application of cold plasma for 2, 5 or 10 min (working gas He/O_2_) did not visibly affect the coloring of wine. However, when the duration of the process was prolonged, the value of this parameter increased from 0.19 to 3.04. The highest values of ∆E* were recorded for wine samples preserved using the combined method (5 min, He/N_2_)—8.72, and the traditional method (addition of 100 mg/L potassium metabisulfite)—8.42. After storage, the lowest ∆E* values were recorded for the wine samples preserved with the combined method (helium/nitrogen; 5 min), ∆E* of 1.01, and cold plasma treatment (helium/nitrogen; 5 min), ∆E* of 1.12. However, the change in color, for most treatments, was imperceptible (∆E* ≤ 3.0) or almost imperceptible (∆E* ≤ 6.0) to the human eye, with the exception of the combined method (2 min, He/O_2_) and the conventional method: ∆E* of 11.81 and 10.98, respectively. 

#### 2.1.2. Polyphenolic Content

The total contents of phenolic compounds and anthocyanins were determined using the spectrophotometric method (Table 2). Measurement of TPC immediately after the application of the different preservation processes did not show considerable differences among the methods used. Most samples showed a slight increase in the content of these compounds compared to the control. The highest concentration of TPC was recorded in the sample with the addition of 100 mg/L potassium metabisulfite and was 6.39% higher than in the unpreserved sample. These results contrast with the data obtained by Lukić et al. (2019), who treated a Cabernet Sauvignon red wine with cold plasma and reported a reduction in phenolic compounds from 1816.06 mg GAE/L to 1606.57 mg GAE/L immediately after 10 min exposure to CP. However, those authors used different process parameters in their study (a different type of plasma generator and working gas), which may have led to differences in the results obtained. Numerous literature data on the effects of cold plasma on the physicochemical properties of food products highlight differences in CP efficiency depending on the process parameters used [28]. The results we obtained after three months of storage showed that the content of TPC in each of the samples was reduced in comparison with the control sample (from 20 to 33% degradation). There are limited reports in the literature explaining the mechanism of action of cold plasma on food products [29,30]. In the generation of CP, emission of light occurs, a shock wave is generated, cavitation processes can develop and, most importantly, free radicals are generated. All this can contribute directly to the degradation of many organic compounds [31]. The highest degree of TPC degradation was obtained in the sample subjected to 10 min of cold plasma treatment (He/O_2_), which is consistent with our previous research findings [2]. Cold plasma generated using a mixture of helium and oxygen as the working gas contains many reactive compounds formed from oxygen, such as hydroxyl radical, peroxyl anion or singlet oxygen, which directly interact with the chemical compounds in the food matrix and can cause their degradation. This topic has been discussed more extensively in recent studies [2,32]. However, comparing the content of phenolic compounds after 3-month storage, we noted in the cold-plasma-preserved sample (He/O_2_; 5 min) a lower content of these compounds by 0.51% compared to the unpreserved stored sample. Furthermore, the concentration of TPC was 1.62% higher compared to the conventionally preserved sample.

Similar relationships were noted when we investigated the total anthocyanin content. However, in contrast to the TPC results, the number of anthocyanins decreased immediately after application of the preservation agent and degraded even more after time. An analysis of the effect of process duration on the content of anthocyanins after 3-month storage showed no clear relationship between increasing process duration and a decrease in the content of these compounds in the samples. Such a trend was observed only for methods in which cold plasma with the working gas He/O_2_ was used as the sterilizing agent. In the case of cold plasma (He/N_2_) or combined treatments, the content of these compounds was higher after 5 min of application of the sterilizing agent compared to 2 min. Similar correlations were also noted in our previous publication, in which we qualitatively and quantitatively determined the content of individual phenolic compounds in the same wine samples using UPLC/MS/MS [2]. Furthermore, when analyzing the anthocyanin content of the stored samples, we clearly observed their higher content in the samples with potassium metabisulfite. In the conventionally preserved samples (addition of potassium metabisulfite at 30 mg/L or 100 mg/L), the anthocyanin content was lower by 4.77% and 3.07%, respectively, compared to the non-preserved stored sample. This is probably related to the protective effect of sulphur dioxide on the enzymatic and non-enzymatic oxidation of wines [33]. 

#### 2.1.3. Antioxidant Activity

Wine has health-promoting properties due to a high content of phenolic compounds, which gives it a strong antioxidant capacity. In the present study, we used DPPH, ABTS and FRAP methods to determine the effect of the different preservation processes studied and three-month storage on the antioxidant capacity of wine.

The results of these assays are presented in Table 2. The findings obtained using the different analytical methods did not always have the same correlations associated with the preservation method used, which may be due to the distinct interactions of the bioactive compounds with the reagents used in the assay. Nevertheless, across all assays, the highest antioxidant activity was recorded in the unstored sample with the addition of 100 mg/L potassium metabisulfite, with DPPH, ABTS and FRAP values of 77.31%, 95.56% and 12.41 mmTE/L, respectively. Interestingly, as discussed above, this sample also had the highest TPC and TAC contents. However, the sample preserved by cold plasma (10 min; He/O_2_) and then stored had the lowest antioxidant capacity. DPPH, ABTS and FRAP values for this sample were 67.03%, 2.83% and 30.82% lower, respectively, compared to the control sample. As in the case of the total content of phenolic compounds and anthocyanins, the effect of the duration of the preservation process on antioxidant activity is not clear. There are no literature data on the direct effect of cold plasma on the antioxidant properties of wine. However, it was shown that exposure of blueberry juice to CP resulted in a decrease in its oxidative potential with increasing treatment time [28]. In contrast, in our study, the radical scavenging activity determined by DPPH was 19% higher after 5 min CP treatment (He/N_2_) compared to 2 min cold plasma exposure. The present results indicate that antioxidant activity is strongly related to the content of phenolic compounds. Similar conclusions have also been reached by other authors who have studied the antioxidant properties of wine [34,35].

### 2.2. Microbial Contamination

Microbiological contamination of food products not only causes earlier spoilage of food, but can also be a threat to consumers’ health. In addition, a shortened shelf life of products generates large amounts of food waste, which is a serious problem in the face of world hunger. Therefore, scientists are still looking for effective food preservation methods. This paper presents a quantitative analysis of the microbiological contamination of wine samples subjected to different preservation methods immediately after the application of a sterilizing agent, as well as after a three-month storage period. In Figure 1A, showing the degree of reduction in microorganisms in the samples that were not subjected to storage, we can see that each of the preservation methods resulted in a reduction of the total number of mesophilic bacteria, and when the combined method (10 min, He/O_2_) was used, no microorganisms were detected in the sample. In addition, an analysis of the graph indicates that the inactivation efficiency of cold plasma increased with increasing process duration. A similar relationship has been reported by numerous authors [18,36,37,38]. For example, Pankaj et al. (2017), who studied the effect of cold plasma (DBD; working gas: air; treatment time: 1–4 min.) on white grape juice reported a reduction in *Saccharomyces cerevisiae* by 7.4 log CFU/mL after 4 min of CP treatment. When we analyzed our samples after 3 months of storage (Figure 1B), we noted a growth in the number of microorganisms in each sample; however, the number of microorganisms was also the lowest in the sample treated for 10 min by the combined method using He/O_2_ as the working gas. The microorganism reduction rate in this sample was 4.21 log number of cfu compared to a stored sample that had not been subjected to any preservation treatment. It should also be noted that this method showed a higher microbial elimination efficiency than the addition of 100 mg/L potassium metabisulfite. The inactivating effect of cold plasma on microorganisms has already been well-documented by numerous authors. Both its direct effect on microbial cells [39,40] and its contribution to the microbiological safety of food products have been analyzed [41,42,43]. The findings available in the literature correspond with ours and indicate that the effectiveness of cold plasma treatment depends on the numerous parameters of the process as well as the food matrix that is subjected to the preservation process.

Biological contamination of wines poses a serious problem not only because it can cause spoilage and reduce the shelf life of these products, but also because the microorganisms present in wine may increase the content of biogenic amines [44]. In our previous publication, we determined the biogenic amine content in the same samples we used in the present experiments [2]. A comparison of those results with the number of microorganisms determined in this study suggests that the content of microorganisms may correlate with the concentration of biogenic amines in wine. For example, the lowest total contents of biogenic amines were recorded in samples 11 and 14 (925.82 μg/L and 955.63 μg/L, respectively), in which the total number of mesophilic bacteria was the lowest in this study. 

### 2.3. Chemometric Analysis

The major goal of the multivariate statistical analysis of the experimental data was to reveal patterns of similarity between the objects or between the variables; to identify specific descriptors responsible for the partitioning of the wine samples; and to elucidate the dataset structure by finding the optimal number of latent variables (factors) able to explain the maximal amount of explained variance of the system. 

The following chemometric methods were used in the intelligent data analysis: Cluster analysis.Factor analysis and principal components analysis.

#### 2.3.1. Cluster Analysis 

Figure 2 shows a hierarchical dendrogram for the clustering of the 18 variables studied (input data standardized by z-transform, squared Euclidean distances as a similarity measure and Ward’s method of linkage). Three major clusters (significance level 1/3Dmax) were identified as follows:

C1: pH, TP, TA, DPPH, ABTS, FRAP—cluster 1 indicates the impact of the acidity, phenolic composition and antioxidant activity of the samples (oxidation factor);

C2: L*, a*, b*, C*, H*, 2-PE—cluster 2 is almost entirely composed of color characteristics; the linkage of 2-PE to the color estimates is surprising to some extent but, in general, cluster 2 reflects color impact as an important descriptor of the investigated wine;

C3: LogN, HIS, PUT, TRP, CAD, TYR—cluster 3 is a representation of the linkage between the microbiological parameter and the biogenic amines (biological factor).

The application of non-hierarchical clustering (K-means) for the a priori selected number of three clusters confirmed the partitioning obtained by hierarchical cluster analysis. Appendix A shows the members of each cluster identified using the K-means clustering approach. The only (insignificant) difference between the clusters obtained using the K-means algorithm versus hierarchical clustering was their numbering (clusters 1 and 2 were formed in reverse order). 

In Figure 3A, the average values for each identified cluster of variables are presented for each of the 30 wine samples. Four different groups of objects are visible in Figure 3A.

The first group includes samples with numbers 1–7, the second 8–15, the third 16–22 and the last, the fourth group—samples 23–30. In general, the first group of samples is characterized by low levels of the “color” factor, the second one by low levels of bioamines (biological impact), the third one by low levels of oxidation impact and the fourth one by high levels of the “color” factor. Additionally, it can be stated that the first group are plasma-preserved samples; the second group are samples preserved with plasma and metabisulfite; the third group are samples treated with plasma after storage; and the fourth group are samples treated with plasma and metabisulfite after storage. It could be assumed from this part of the statistical analysis that the time of treatment and the concentration of the metabisulfite added were not significant factors.

Appendix A shows the non-hierarchical partitioning of the 30 wine samples (objects), which reveals patterns of similarity between the objects described by all the variables. A preliminary hypothesis assumed the existence of four clusters depending on the conditions of preservation (plasma or plasma plus metabisulfite) and storage (without storage and after storage).

It is of substantial interest to see which variables (descriptors) are specific for the partitioning of the group of objects into four clusters.

Figure 3B shows the average values of each variable for each of the identified clusters.

Cluster 1 is characterized by the lowest levels of color parameters (all samples preserved by plasma and metabisulfite and stored afterwards). Another feature of this group of samples is the lowest levels of antioxidant parameters and moderate levels of biogenic amines. Very specific for this cluster is the highest level of log N and the relatively high level of 2-PE.

Cluster 2 is the largest one (it mainly contains samples preserved with plasma, with and without storage). It is characterized by moderate levels of almost all the variables and the highest levels of biogenic amines. 

The other two clusters are relatively small (cluster 3, with the highest levels of color descriptors, chiefly contains non-preserved control samples and cluster 4 mainly consists of samples preserved using helium/oxygen), characterized by high levels of antioxidants and low levels of biogenic amines. 

The graphical relation between the objects and the variables is illustrated by the two-way clustering diagram in Figure 4A.

The dark brown regions represent strong relations (e.g., biogenic amines are strongly associated with objects 2, 17, 20, 4, 6 and 22), and green regions indicate a lack of a strong relation (e.g., TA are weakly associated with objects 17, 22, 14, 27, 29 and 23).

#### 2.3.2. Factor Analysis and Principal Component Analysis

Appendix A shows the factor loadings for the 18 variables studied.

Three latent factors explain over 80% of the total variance of the system. In general, the grouping by high factor loadings in the different latent components corresponds to the results obtained by cluster analysis. The first latent factor, which explains over 30% of the total variance, could be provisionally named an “antioxidant” factor due to the significant loadings of the phenolic compounds, pH and antioxidants. The negative sign of logN indicates another level of the relationship.

The second latent factor explains nearly 30% of the total variance and includes high loadings for biogenic amines, which is why it can be tentatively referred to as a “biogenic amines factor”. The only exception is 2-PE (negative sign of the loading), but this could be explained by the low variability of the variable (it takes only two values for all objects).

The third latent factor could be conditionally called a “color” factor, as it includes all the color indicators (it explains nearly 30% of the total variance).

Figure 4B shows a 2D plot of the factor loadings. It very clearly illustrates the formation of the three latent factors and the special positions of logN and 2-PE.

## 3. Materials and Methods

### 3.1. Wine

In this study, we investigated a red wine produced at the Dom Bliskowice winery located in Poland’s Lubelskie Province. Two grape varieties, Rondo and Regent (1:1), obtained from the October 2019 harvest, were used to produce the wine. The test samples included wine subjected to different preservation processes: cold plasma treatment, preservation with the addition of 30 mg/L or 100 mg/L potassium metabisulfite and a method combining cold plasma with the addition of 30 mg/L potassium metabisulfite. The control sample was wine not subjected to any preservation process. Samples were assayed immediately after preservation and again after three months of storage. 

### 3.2. Cold Plasma Treatment

50 mL of wine contained in a sterile glass container was placed on a magnetic stirrer to ensure that the plasma generated in the Dielectric Barrier Discharge (DBD) reactor was uniformly applied to the samples. A mixture of helium and nitrogen or helium and oxygen was used as the working gas. The preservation process was carried out for 2, 5 or 10 min. The exact methodology for cold plasma treatment was described in an earlier publication by Niedźwiedź et al. (2022). 

### 3.3. Determination of pH and Color Measurement

The pH value of the red wine samples was measured potentiometrically using a Hanna HI 221 pH meter (Hanna Instruments, Woonsocket, RI, USA). The electrode tip was thoroughly rinsed each time with distilled water both before and after measurement. 

Color parameters were determined with an X-Rite 8200 colorimeter (X-Rite, Inc., Grand Rapids, MI, USA) using the CIELab color space (Method OIV-MA-AS2-11, 2006). Black slides and white plates were used to calibrate the colorimeter. The spectra were registered directly on the wine, using a 10 mm optical path glass cell. Color was expressed by the CIE coordinates: L*—clarity, lightness; a*—red/green color components; and b*—blue/yellow color components, and by its derived magnitudes: C*—chroma and H*—hue angle. In order to determine the degree of color change between the study samples and the control, delta E (∆E*) was calculated according to the following formula:(1)$$\Delta E*=\sqrt{\left(\Delta L^{2}+\Delta a^{2}+\Delta b^{2}\right)}$$
where: ∆***L***; ∆***a***; and ∆***b*** are the difference between the value of a given parameter of the test sample and the control sample.

### 3.4. Determination of Polyphenolic Compounds

#### 3.4.1. Total Phenolic Content

The total phenolic content (TPC) of red wine was determined by the Folin–Ciocalteu method with a minor modification. Wine samples were diluted 1:9 with distilled water. Fifty microliters (50 μL) of diluted sample was mixed with 750 μL of Folin–Ciocalteu reagent. After 5 min, 750 μL of sodium hydrogen carbonate (NaHCO_3_) was added to the mixture followed by incubation for 2 h at room temperature. The absorbance was measured at a wavelength of 760 nm against the blank sample. Results were expressed as mg/L gallic acid equivalents (mg GAE/L).

#### 3.4.2. Total Anthocyanin Content 

For the determination of the total anthocyanin content (TAC), two reaction solutions were prepared by mixing (1) 50 μL of diluted wine sample (10×) with 200 μL of KCl buffer (pH 1.0) and (2) 50 μL of diluted wine sample (10×) with 200 μL of CH_3_COONa × 3 H_2_O buffer (pH 4.5). Subsequently, the samples were incubated for 15 min. After that time, the absorbance was measured at two wavelengths: 520 nm and 700 nm. TAC was calculated using the formulas below, and the results were expressed as cyanidin-3-glucoside equivalents.
(2)$$A={\left(A_{520}-A_{700}\right)}_{pH\;1.0}-{\left(A_{520}-A_{700}\right)}_{pH\;4.5}$$
(3)$$C=\frac{\left(A\times MW\times DF\times 1000\right)}{\left(\varepsilon \times 1\right)}$$
where *A*—absorbance of sample; *C*—anthocyanin concentration; *MW*—molar mass of cyanidin-3-glucoside; *DF*—sample dilution; *ε*—cyanidin-3 molar extinction coefficient of glucoside; and 1—length of light path.

### 3.5. Determination of Antioxidant Activity

#### 3.5.1. DPPH Inhibition

Antioxidant activity was determined using the DPPH radical according to a slightly modified method proposed by Brand-Williams et al. (1995) [45] and Szwajgier et al. (2021) [46]. A 10-fold diluted test sample in a volume of 20 μL was mixed with 255 μL of methanol and 30 μL of a DPPH solution with a fixed absorbance (1.5 at 515 nm). A blank sample was prepared in an identical manner by replacing the diluted wine sample with 20 μL of deionized water. Then, after 4 min of incubation, the absorbance of the samples was measured at 515 nm using a microplate reader. The experiment was performed in three replicates, and the results were expressed as % inhibition of DPPH including standard deviations. The calculation was performed according to the following equation:(4)$$\%DPPHinhibition=(A_{B}-A_{S})/A_{B\;}\times 100,$$
where *A_B_*—absorbance of the blank sample, *A_S_*—absorbance of the test sample. 

#### 3.5.2. ABTS

Antioxidant activity was assayed using the radical cation ABTS according to Miller et al. (1993) [47] with modifications. For analysis, the test sample diluted 10-fold in a volume of 20 μL was mixed with 180 μL of distilled water and 185 μL of ABTS solution (absorbance = 1.5 at 734 nm). After 4 min, absorbance was measured. The negative blank sample contained 0.2 mL of distilled water and 0.185 μL of ABTS solution. Sample background was subtracted during calculations. All samples were analyzed in three replicates, and the results were expressed as % inhibition of ABTS including standard deviations

#### 3.5.3. FRAP

Antioxidant activity was determined based on the degree of ferric ion reduction using the FRAP method according to the procedure described by Szwajgier et al. (2021). A FRAP solution was prepared for the analysis by mixing 2.5 mL of a 5 mM 2,4,6-tris(2-pyridyl)-(S)-triazine (TPTZ) solution (in a 40 mM HCl solution), 2.5 mL of a 5 mM FeCl3 solution and 25 mL of acetate buffer (0.3 M pH 3.6). The mixture was then heated in a water bath at 37 °C for 20 min. After reagent preparation, 20 ul of a wine sample was mixed with 1.9 mL of the FRAP solution and shaken for 30 min at room temperature. Absorbance was measured at 593 nm after shaking. Results were expressed as Trolox equivalents (mg Trolox/mL).

### 3.6. Microbiological Analysis

The microbiological purity of the wine samples was evaluated using a pour plate method. For this purpose, a series of ten-fold dilutions was first prepared by mixing 1 mL of wine with 9 mL of saline solution. Then, 1 mL of the appropriately diluted solution was applied to the center of a sterile Petri dish, and 15 mL of sterile, cooled nutrient agar medium (BTL, Łódź, Poland) was poured over it. The medium with the test sample was mixed well and incubated for 72  h at 30 °C. After incubation, colonies were counted, and the number of viable cells was determined as the mean of log colony forming units (cfu) per mL of sample ± standard deviation. 

### 3.7. Chemometric Analysis

Multivariate statistical data mining was used to discover specific correlations between the different wine preservation methods and their effects on the physicochemical and biological properties of wine. The chemometric methods used included cluster analysis (hierarchical and non-hierarchical) and factor analysis. The input dataset consisted of 30 objects (wine samples subjected to different preservation methods) described by 18 variables that could be divided into 6 categories: pH, color, microbiological analysis, polyphenolic compounds, antioxidant activity—which were determined in this study—and biogenic amines, which had been determined in an earlier work (Niedźwiedź et al., 2022). Statistical analysis was conducted using STATISTICA 8.0 software (New York, NY, USA).

In addition, all data obtained were expressed as mean ± standard deviation (*n* ≥ 3). Differences between mean data values were tested for statistical significance at *p* < 0.05 using analysis of variance and Tuckey’s test.

## 4. Conclusions

The physicochemical and biological properties of wine play a key role in the consumer’s assessment of its quality. In the present study, we investigated the effect of the use of cold plasma as a preservation method on the physicochemical and biological properties of red wine and compared it with the effects of using a conventional preservation method (addition of 30 or 100 mg/L of potassium metabisulfite) and a combined method (cold plasma with 30 mg/L of potassium metabisulfite). In addition, the effect of storage time after the application of each of the analyzed methods was assessed to determine the potential of the respective techniques to extend the shelf life of the product. 

More specifically, we analyzed the effects of the different preservation methods on the pH; color; total content of phenolic compounds and anthocyanins; and antioxidant activity, as well as the biological safety of wine. Color is a parameter that plays a decisive role in the evaluation of wine quality. We examined changes in wine color under the influence of preservative factors after three-month storage using the CIELab space. The least prominent color changes were observed in samples treated with the combined method (helium/nitrogen; 5 min), ∆E*—1.01, and cold plasma (helium/nitrogen; 5 min), ∆E*—1.12. By contrast, the most perceptible changes were noted in samples preserved with the traditional method: ∆E*—11.81. In addition, a reduction in the content of phenolic compounds, and thus a decrease in antioxidant activity, was observed in stored samples. This effect was the mildest for preservation methods involving the addition of potassium metabisulfite and those that used a mixture of helium and nitrogen as the working gas. These results correspond with the data we obtained in a previous publication. When analyzing the effect of the selected preservation methods on the biological purity of the wine, we observed a lower number of microorganisms in the methods where cold plasma was used. After 3 months of storage, the total content of microorganisms in the samples was lower by 4.21 and 3.17 log(N), for the sample treated for 10 min with the combined method using He/O_2_ and He/N_2_, respectively, compared to the sample stored unpreserved. The above results suggest that the action of cold plasma using a mixture of helium and nitrogen as the working gas has a smaller impact on the final quality of the wine. However, the impact of the process duration on individual parameters is ambiguous and requires further detailed studies.

Obtained data allow us to assume that, in the future, cold plasma may contribute to the reduction or elimination of the use of SO_2_ in the wine industry. However, since still relatively little is known about the influence of cold plasma on wine, it is necessary to conduct further research in order to be able to fully exploit the potential of this technology on an industrial scale in the future.

## Figures and Tables

**Figure 1 molecules-27-07048-f001:**
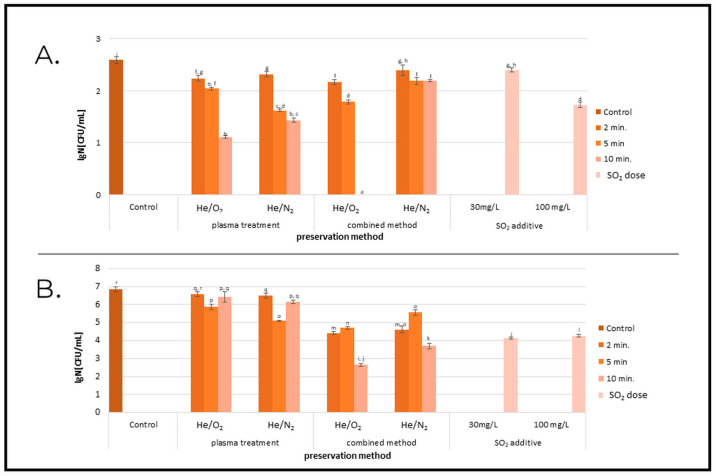
Effect of inactivation of bacterial growth at different preservation methods. (**A**)—before storage. (**B**)—after storage. Cold plasma exposure time—2, 5, or 10 min. Working gas was a mixture of He/O_2_ or He/N_2_; ^a–r^ Values with the different superscript letters are significantly different (*p* < 0.05).

**Figure 2 molecules-27-07048-f002:**
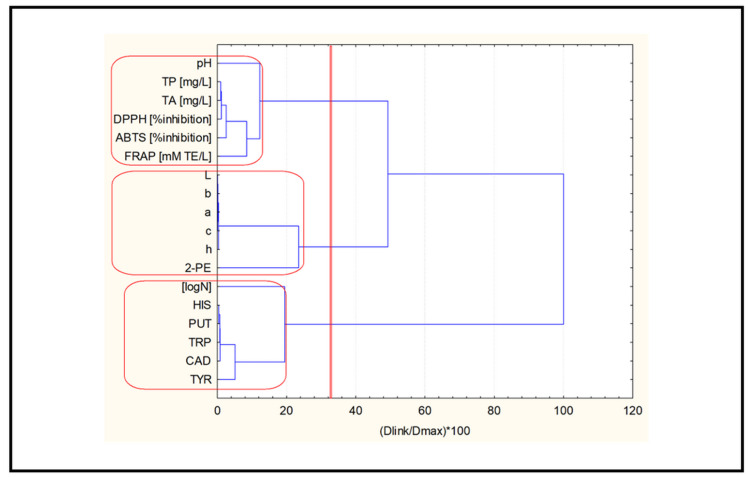
Hierarchical dendrogram for linkage of 18 variables. TRP—tryptamine; PUT—putrescine; HIS—histamine; TYR—tyramine; CAD—cadaverine; 2-PE—2-phenylethylamine.

**Figure 3 molecules-27-07048-f003:**
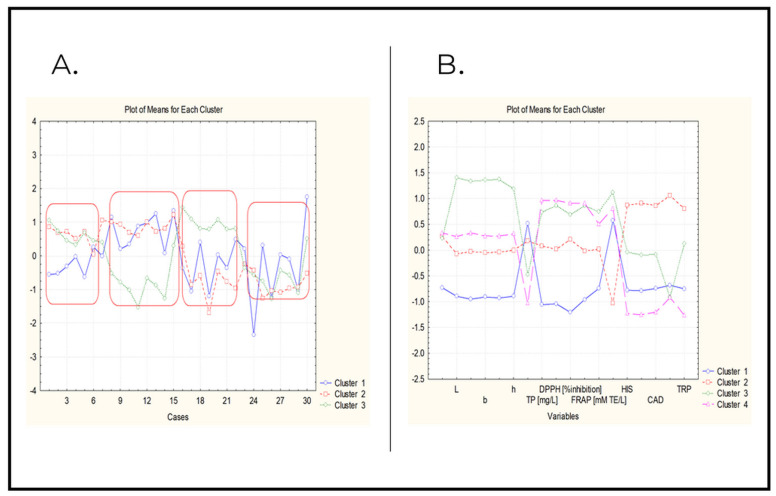
(**A**) Plot of average values for each cluster of variables for each object wine sample. (**B**) Plot of average values of each variable for each identified cluster. The sequence of variables is as follows: pH, L*, a*, b*, C*, H*, logN, TP, TA, DDPH, ABTS, FRAP, 2-PE, HIS, PUT, CAD, TYR, TRP. TRP—tryptamine; PUT—putrescine; HIS—histamine; TYR—tyramine; CAD—cadaverine; 2-PE—2-phenylethylamine.

**Figure 4 molecules-27-07048-f004:**
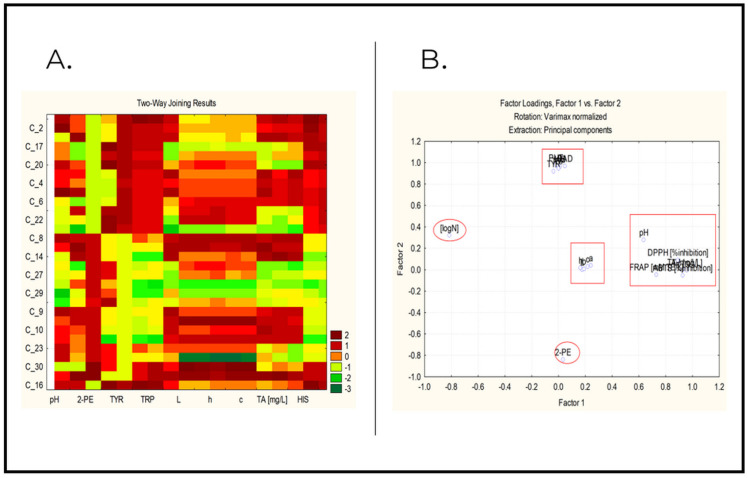
(**A**)—Two-way joining plot. (**B**)—2D plot of factor loadings. TRP—tryptamine; PUT—putrescine; HIS—histamine; TYR—tyramine; CAD—cadaverine; 2-PE—2-phenylethylamine.

**Table 1 molecules-27-07048-t001:** Effect of different preservation processes and storage time on pH and color of red wine.

Sample	Preservation Methods	pH	L*	a*	b*	C*	H*	∆E*
BEFORE STORAGE								
1.	no preservation	3.54 ± 0.01 ^de^	11.23 ± 0.02 ^e^	41.68 ± 0.08 ^ef^	19.35 ± 0.07 ^g^	45.96 ± 0.13 ^ef^	24.91 ± 0.03 ^e^	-
2.	cold plasma (2 min *; He/O_2_ **)	3.55 ± 0.02 ^e^	11.28 ± 0.03 ^ef^	41.84 ± 0.22 ^ef^	19.45 ± 0.11 ^g^	46.14 ± 0.05 ^fg^	24.93 ± 0.07 ^e^	0.20 ± 0.01
3.	cold plasma (5 min; He/O_2_)	3.52 ± 0.01 ^bcde^	11.81 ± 0.07 ^g^	42.43 ± 0.07 ^fghij^	20.36 ± 0.02 ^h^	47.06 ± 0.07 ^i^	25.64 ± 0.12 ^f^	1.39 ± 0.09
4.	cold plasma (10 min; He/O_2_)	3.52 ± 0.01 ^bcde^	12.56 ± 0.06 ^ij^	43.16 ± 0.17 ^jk^	21.65 ± 0.22 ^l^	48.29 ± 0.06 ^jk^	26.64 ± 0.09 ^ij^	3.04 ± 0.12
5.	cold plasma (2 min; He/N_2_)	3.48 ± 0.01 ^abcd^	11.02 ± 0.05 ^e^	41.51 ± 0.16 ^e^	19.01 ± 0.03 ^f^	45.65 ± 0.0 ^e^	24.61 ± 0.10 ^e^	0.43 ± 0.02
6.	cold plasma (5 min; He/N_2_)	3.49 ± 0.01 ^abcde^	13.26 ± 0.22 ^k^	44.01 ± 0.12 ^lmn^	22.87 ± 0.10 ^m^	49.60 ± 0.23 ^l^	27.46 ± 0.06 ^m^	4.68 ± 0.16
7.	cold plasma (10 min; He/N_2_)	3.55 ± 0.02 ^e^	12.55 ± 0.52 ^ij^	43.29 ± 0.37 ^kl^	21.70 ± 0.10 ^l^	48.41 ± 0.19 ^jk^	26.62 ± 0.03 ^ij^	3.14 ± 0.09
8.	30 mg/L potassium metabisulfite	3.54 ± 0.01 ^de^	14.79 ± 012 ^n^	45.89 ± 0.04 ^qr^	25.24 ± 0.10 ^r^	52.37 ± 0.33 ^p^	28.82 ± 0.20 ^pq^	8.07 ± 0.22
9.	cold plasma (2 min; He/O_2_) and 30 mg/L potassium metabisulfite	3.54 ± 0.01 ^de^	12.31 ± 0.10 ^hij^	43.05 ± 0.25 ^ijk^	21.23 ± 0.14 ^jk^	48.00 ± 0.14 ^j^	26.24 ± 0.15 ^h^	2.56 ± 0.18
10.	cold plasma (5 min; He/O_2_) and 30 mg/L potassium metabisulfite	3.52 ± 0.00 ^bcde^	12.69 ± 0.14 ^j^	43.44 ± 1.27 ^klm^	21.87 ± 0.05 ^l^	48.64 ± 0.05 ^k^	26.73 ± 0.17 ^jk^	3.40 ± 0.10
11.	cold plasma (10 min; He/O_2_) and 30 mg/L potassium metabisulfite	3.52 ± 0.01 ^bcde^	14.10 ± 0.06 ^m^	44.88 ± 0.12 ^op^	23.98 ± 0.14 ^o^	51.04 ± 0.14 ^n^	28.51 ± 0.12 ^op^	6.32 ± 0.13
12.	cold plasma (2 min; He/N_2_) and 30 mg/L potassium metabisulfite	3.52 ± 0.02 ^bcde^	14.33 ± 0.08 ^m^	45.27 ± 0.11 ^pq^	24.62 ± 0.10 ^p^	51.54 ± 0.16 ^o^	28.54 ± 0.18 ^op^	7.09 ± 0.09
13.	cold plasma (5 min; He/N_2_) and 30 mg/L potassium metabisulfite	3.51 ± 0.01 ^bcde^	15.13 ± 0.14 ^n^	46.08 ± 0.22 ^qr^	25.80 ± 0.16 ^r^	52.81 ± 0.12 ^q^	29.24 ± 0.13 ^rs^	8.73 ± 0.18
14.	cold plasma (10 min; He/N_2_) and 30 mg/L potassium metabisulfite	3.48 ± 0.01 ^abcd^	12.13 ± 0.15 ^ghi^	42.75 ± 0.15 ^ghijk^	20.91 ± 0.07 ^i^	46.92 ± 0.11 ^i^	26.06 ± 0.13 ^gh^	2.09 ± 0.03
15.	100 mg/L potassium metabisulfite	3.52 ± 0.02 ^bcde^	14.94 ± 0.06 ^n^	46.16 ± 0.09 ^r^	25.44 ± 0.12 ^q^	52.79 ± 0.13 ^pq^	29.03 ± 0.12 ^qr^	8.42 ± 0.01
AFTER STORAGE								
16.	no preservation	3.53 ± 0.01 ^cde^	11.71 ± 0.15 ^fg^	42.22 ± 0.04 ^efghi^	20.19 ± 0.14 ^h^	46.80 ± 0.08 ^hi^	25.55 ± 0.10 ^f^	1.11 ± 0.12
17.	cold plasma (2 min; He/O_2_)	3.50 ± 0.02 ^abcde^	10.18 ± 0.15 ^d^	39.55 ± 0.05 ^cd^	17.56 ± 0.11 ^e^	43.27 ± 0.14 ^d^	23.93 ± 0.12 ^d^	2.97 ± 0.09
18.	cold plasma (5 min; He/O_2_)	3.48 ± 0.01 ^abcd^	13.62 ± 0.09 ^kl^	44.21 ± 0.07 ^mno^	23.48 ± 0.09 ^n^	50.05 ± 0.08 ^m^	27.97 ± 0.10 ^n^	5.40 ± 0.08
19.	cold plasma (10 min; He/O_2_)	3.47 ± 0.01 ^abc^	9.72 ± 0.09 ^c^	39.89 ± 0.12 ^d^	16.75 ± 0.09 ^d^	43.26 ± 0.12 ^d^	22.80 ± 0.09 ^c^	3.50 ± 0.10
20.	cold plasma (2 min; He/N_2_)	3.53 ± 0.02 ^cde^	12.74 ± 0.08 ^j^	42.99 ± 0.11 ^ijk^	21.96 ± 0.09 ^l^	48.28 ± 0.10 ^jk^	27.05 ± 0.10 ^kl^	3.29 ± 0.10
21.	cold plasma (5 min; He/N_2_)	3.50 ± 0.01 ^abcde^	11.74 ± 0.10 ^g^	42.13 ± 0.15 ^efgh^	20.24 ± 0.14 ^h^	46.74 ± 0.11 ^hi^	25.67 ± 0.10 ^f^	1.12 ± 0.12
22.	cold plasma (10 min; He/N_2_)	3.48 ± 0.01 ^abcd^	13.91 ± 0.12 ^lm^	44.34 ± 0.13 ^no^	23.98 ± 0.12 ^o^	50.41 ± 0.06 ^m^	22.41 ± 0.09 ^o^	5.97 ± 0.08
23.	30 mg/L potassium metabisulfite	3.53 ± 0.01 ^cde^	12.35 ± 0.06 ^hij^	43.05 ± 0.11 ^jk^	21.30 ± 0.11 ^k^	48.03 ± 0.14 ^j^	26.32 ± 0.08 ^hi^	2.63 ± 0.09
24.	cold plasma (2 min; He/O_2_) and 30 mg/L potassium metabisulfite	3.53 ± 0.02 ^cde^	6.6 ± 0.08 ^a^	34.28 ± 0.14 ^a^	11.39 ± 0.03 ^a^	36.13 ± 0.04 ^a^	18.37 ± 0.04 ^a^	11.81 ± 0.07
25.	cold plasma (5 min; He/O_2_) and 30 mg/L potassium metabisulfite	3.49 ± 0.01 ^abcde^	12.73 ± 0.05 ^j^	42.76 ± 0.06 ^hijk^	21.97 ± 0.06 ^l^	48.07 ± 0.07 ^j^	27.18 ± 0.04 ^lm^	3.21 ± 0.06
26.	cold plasma (10 min; He/O_2_) and 30 mg/L potassium metabisulfite	3.47 ± 0.01 ^abc^	8.94 ± 0.05 ^b^	38.39 ± 0.07 ^b^	15.41 ± 0.11 ^b^	41.37 ± 0.04 ^b^	21.87 ± 0.07 ^b^	5.62 ± 0.09
27.	cold plasma (2 min; He/N_2_) and 30 mg/L potassium metabisulfite	3.48 ± 0.00 ^abcd^	12.03 ± 0.03 ^gh^	41.92 ± 0.06 ^efg^	20.73 ± 0.12 ^i^	46.77 ± 0.09 ^hi^	26.32 ± 0.04 ^hi^	1.61 ± 0.09
28.	cold plasma (5 min; He/N_2_) and 30 mg/L potassium metabisulfite	3.46 ± 0.01 ^ab^	11.73 ± 0.08 ^g^	41.79 ± 0.05 ^ef^	20.22 ± 0.06 ^h^	46.43 ± 0.09 ^gh^	25.82 ± 0.05 ^fg^	1.01 ± 0.07
29.	cold plasma (10 min; He/N_2_) and 30 mg/L potassium metabisulfite	3.44 ± 0.03 ^a^	9.44 ± 0.06 ^c^	38.94 ± 0.06 ^bc^	16.28 ± 0.06 ^c^	42.21 ± 0.07 ^c^	22.69 ± 0.07 ^c^	4.49 ± 0.06
30.	100 mg/L potassium metabisulfite	3.47 ± 0.01 ^abc^	16.52 ± 0.12 ^o^	47.60 ± 0.09 ^s^	26.94 ± 0.03 ^s^	54.70 ± 0.05 ^r^	29.51 ± 0.07 ^s^	10.98 ± 0.05

^a–s^ Values with the different superscript letters within one column are significantly different (*p* < 0.05). * Cold plasma exposure time—2, 5, or 10 min. ** Working gas (a mixture of He/O_2_ or He/N_2_).

**Table 2 molecules-27-07048-t002:** Effect of different preservation processes and storage time on polyphenolic content and antioxidant activity of red wine.

Sample Number	Preservation Methods	TPC [mg/L]	TAC [mg/L]	DPPH [% Inhibition]	ABTS [% Inhibition]	FRAP [mM TE/L]
BEFORE STORAGE						
1.	no preservation	2442.75 ± 12.30 ^mn^	690.92 ± 3.00 ^lmno^	72.22 ± 0.87 ^hij^	95.24 ± 1.73 ^a^	10.35 ± 0.53 ^fghijk^
2.	cold plasma (2 min *; He/O_2_ **)	2300.25 ± 10.15 ^j^	647.92 ± 3.64 ^ijk^	67.82 ± 1.27 ^h^	95.13 ± 2.54 ^a^	10.16 ± 0.59 ^fghij^
3.	cold plasma (5 min; He/O_2_)	2497.03 ± 11.26 ^pq^	607.00 ± 17.82 ^h^	74.14 ± 1.46 ^jk^	95.39 ± 0.95 ^a^	10.72 ± 0.11 ^hijkl^
4.	cold plasma (10 min; He/O_2_)	2449.53 ± 4.02 ^n^	634.56 ± 3.08 ^hi^	72.22 ± 0.88 ^hij^	94.81 ± 1.40 ^a^	9.81 ± 0.19 ^defghi^
5.	cold plasma (2 min; He/N_2_)	2483.46 ± 5.26 ^op^	693.84 ± 4.98 ^mno^	70.50 ± 1.09 ^hij^	95.22 ± 2.33 ^a^	12.06 ± 0.46 ^lm^
6.	cold plasma (5 min; He/N_2_)	2388.46 ± 3.59 ^k^	642.35 ± 10.51 ^ij^	72.61 ± 4.31 ^ijk^	94.26 ± 1.68 ^a^	8.24 ± 0.27 ^abcd^
7.	cold plasma (10 min; He/N_2_)	2517.39 ± 4.07 ^q^	649.17 ± 3.01 ^ijk^	72.03 ± 0.99 ^hij^	95.39 ± 1.56 ^a^	11.64 ± 0.61 ^jklm^
8.	30 mg/L potassium metabisulfite	2422.39 ± 3.03 ^lm^	702.19 ± 8.22 ^nop^	73.75 ± 1.84 ^ijk^	95.27 ± 1.12 ^a^	11.42 ± 0.65 ^jklm^
9.	cold plasma (2 min; He/O_2_) and 30 mg/L potassium metabisulfite	2381.68 ± 4.80 ^k^	658.35 ± 5.74 ^ijk^	72.61 ± 2.52 ^ijk^	95.42 ± 0.62 ^a^	11.45 ± 0.43 ^jklm^
10.	cold plasma (5 min; He/O_2_) and 30 mg/L potassium metabisulfite	2483.46 ± 5.71 ^op^	670.46 ± 5.75 ^jklm^	69.73 ± 1.51 ^hij^	95.30 ± 1.25 ^a^	10.17 ± 0.24 ^fghij^
11.	cold plasma (10 min; He/O_2_) and 30 mg/L potassium metabisulfite	2463.10 ± 8.96 ^no^	662.67 ± 22.00 ^ijkl^	72.22 ± 2.35 ^hij^	95.04 ± 1.03 ^a^	9.81 ± 0.25 ^defghi^
12.	cold plasma (2 min; He/N_2_) and 30 mg/L potassium metabisulfite	2578.46 ± 9.04 ^s^	673.80 ± 6.31 ^klmn^	70.11 ± 0.97 ^hij^	95.51 ± 0.93 ^a^	12.09 ± 0.76 ^lm^
13.	cold plasma (5 min; He/N_2_) and 30 mg/L potassium metabisulfite	2415.61 ± 4.34 ^l^	658.56 ± 14.13 ^ijk^	69.16 ± 1.08 ^hi^	95.24 ± 0.37 ^a^	11.45 ± 0.69 ^jklm^
14.	cold plasma (10 min; He/N_2_) and 30 mg/L potassium metabisulfite	2551.31 ± 3.46 ^r^	707.20 ± 5.21 ^op^	73.95 ± 0.95 ^jk^	95.42 ± 1.59 ^a^	11.86 ± 0.77 ^klm^
15.	100 mg/L potassium metabisulfite	2598.81 ± 5.18 ^s^	730.16 ± 14.61 ^p^	77.31 ± 1.34 ^k^	95.56 ± 1.55 ^a^	12.41 ± 0.59 ^m^
AFTER STORAGE						
16.	no preservation	1954.20 ± 5.37 ^i^	571.10 ± 1.67 ^g^	57.21 ± 0.89 ^g^	94.80 ± 0.93 ^a^	11.32 ± 0.61 ^ijklm^
17.	cold plasma (2 min; He/O_2_)	1750.64 ± 5.38 ^e^	485.10 ± 7.23 ^cd^	46.66 ± 0.65 ^ef^	93.36 ± 0.56 ^a^	8.17 ± 0.14 ^abc^
18.	cold plasma (5 min; He/O_2_)	1791.35 ± 4.18 ^f^	476.47 ± 12.11 ^cd^	36.13 ± 0.57 ^d^	93.86 ± 1.16 ^a^	8.15 ± 0.46 ^abc^
19.	cold plasma (10 min; He/O_2_)	1587.79 ± 6.28 ^a^	425.40 ± 3.26 ^a^	23.81 ± 0.34 ^a^	92.54 ± 0.51 ^a^	7.16 ± 0.17 ^a^
20.	cold plasma (2 min; He/N_2_)	1832.06 ± 7.03 ^g^	436.95 ± 11.12 ^ab^	31.44 ± 1.70 ^cd^	94.14 ± 1.88 ^a^	10.41 ± 0.56 ^ghijk^
21.	cold plasma (5 min; He/N_2_)	1944.20 ± 5.09 ^i^	529.91 ± 15.49 ^ef^	50.18 ± 1.30 ^f^	94.20 ± 1.44 ^a^	11.46 ± 0.77 ^jklm^
22.	cold plasma (10 min; He/N_2_)	1791.35 ± 7.91 ^f^	436.12 ± 4.89 ^ab^	46.07 ± 1.03 ^ef^	93.20 ± 0.88 ^a^	9.05 ± 0.30 ^bcdefg^
23.	30 mg/L potassium metabisulfite	1832.06 ± 5.95 ^g^	543.83 ± 9.96 ^fg^	31.99 ± 0.67 ^cd^	94.51 ± 0.73 ^a^	10.07 ± 0.42 ^efghij^
24.	cold plasma (2 min; He/O_2_) and 30 mg/L potassium metabisulfite	1750.64 ± 8.50 ^e^	467.57 ± 10.85 ^c^	30.85 ± 0.25 ^bc^	94.20 ± 1.44 ^a^	9.73 ± 0.16 ^cdefghi^
25.	cold plasma (5 min; He/O_2_) and 30 mg/L potassium metabisulfite	1628.50 ± 7.48 ^b^	437.23 ± 5.03 ^ab^	26.16 ± 0.71 ^ab^	93.46 ± 1.74 ^a^	7.82 ± 0.16 ^ab^
26.	cold plasma (10 min; He/O_2_) and 30 mg/L potassium metabisulfite	1709.92 ± 4.34 ^d^	433.54 ± 2.55 ^ab^	32.02 ± 0.98 ^cd^	93.30 ± 1.66 ^a^	8.52 ± 0.36 ^abcde^
27.	cold plasma (2 min; He/N_2_) and 30 mg/L potassium metabisulfite	1669.21 ± 5.93 ^c^	458.38 ± 7.07 ^bc^	35.51 ± 0.52 ^cd^	93.17 ± 1.84 ^a^	9.12 ± 0.42 ^bcdefg^
28.	cold plasma (5 min; He/N_2_) and 30 mg/L potassium metabisulfite	1709.92 ± 2.37 ^d^	482.32 ± 6.27 ^cd^	43.72 ± 0.90 ^e^	93.70 ± 0.66 ^a^	9.19 ± 0.60 ^bcdefgh^
29.	cold plasma (10 min; He/N_2_) and 30 mg/L potassium metabisulfite	1832.06 ± 6.93 ^g^	501.24 ± 4.12 ^de^	44.90 ± 1.65 ^e^	94.20 ± 1.45 ^a^	8.81 ± 0.31 ^bcdef^
30.	100 mg/L potassium metabisulfite	1913.49 ± 11.40 ^h^	553.57 ± 9.45 ^fg^	50.18 ± 2.61 ^f^	94.72 ± 0.56 ^a^	10.31 ± 0.57 ^fghijk^

^a–s^ Values with the different superscript letters within one column are significantly different (*p* < 0.05); * Cold plasma exposure time—2, 5, or 10 min. ** Working gas (a mixture of He/O_2_ or He/N_2_).

## Data Availability

Not applicable.

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
