# Peer review of "Comparison of the Effect of Cold Plasma with Conventional Preservation Methods on Red Wine Quality Using Chemometrics Analysis"

_molecules, 2022, doi:10.3390/molecules27207048_

Round 1

Reviewer 1 Report

The manuscript of Niedźwiedź et al. wanted to study the influence of cold plasma to the quality and the shelf life of red wine. Before being considered for publication in molecules, I believe this manuscript needs major revision.

First of all, if it is for the purpose of extending the shelf life as stated in the title, I think the importance of applying non-thermal technologies such as cold plasma in wine needs to be considered.

The text requires revision of the English language;

Most of the pictures are vague, and some texts overlap seriously, please standardize them.

Table 1 contains a large number of figures and significance, please check carefully. And in my opinion, the authors’ way of significance labeling is wrong. We will label the maximum value as a.

There is a lack of significance and analysis in the picture.

Abstract: The abstract has to be rewritten, I cannot get valid information in the abstract.

Introduction

The description is broad and does not highlight the main point.

Parts of the description were unreasonable. For example, according to the authors, the use of cold plasma to reduce sulfur usage, not heat treatment, while the introduction focuses on the non-thermal properties of cold plasma.

Text

Discussion is superficial, and some arguments lack references;

A vital problem is that from the data listed by the authors, especially from Table 1 and the column charts without significant labels, I cannot clearly draw the significant difference between the treatments, so I’m confused whether the alleged processing is really necessary.

Reviewer 2 Report

Very good paper with detailed analysis. I would like to the authors to change the title and include chemometric analysis.

Reviewer 3 Report

General comments:

Being the follow up and a supplementary study to a previous publication of the same research group (Niedźwiedź et al., 2022), the main scope of this manuscript, according to authors is ‘to analyze the effects of the CP treated and a combined method, as well as the effect of storage time on the physicochemical properties (pH, color, total polyphenol con-94 tent, total anthocyanin content, antioxidant properties) and biosafety of red wine’. The subject is really interesting and is worthy studying. However, there are numerous points that need a major reconsideration, much more in depth explanations, in order to provide a clear and well sustained conclusion. The ‘Introduction’ and the ‘Materials and Methods’ parts are well organized and provide the essential information on Cold Plasma (CP) effects and advantages, as well as the methods followed for the experimental part. Nonetheless, in my opinion, the experimental design of this research work is inadequate, as far as the storage study is concerned. Red wine, being a product of extended shelf life, should be measured for a much longer period than the 3 months storage, to really assess the impact of the different preservation methods presented here. Additionally, more sensory attributes should be measured, immediately after the process (zero time for storage), as well as during storage. In recent literature, is there any information on the impact of CP on sensory attributes of other foods/drinks? Since you mention the negative effect of potassium metabisulfite (60-62), it is necessary to be able to assess the sensory profile of a CP treated wine, and especially its aroma/taste characteristics during subsequent storage.

Another point refers to the difficulty to compare the effectiveness of the different preservation methods, using Table 1. Since the aim of this work is not only to assess the impact immediately after the process, but also the stability of wine under storage, it would help to provide the % change of the numerous indices measured, for the differently treated samples. In the way numbers are shown, it is hard to understand which treatment delivered the more stable products, in terms of each parameter measured.

Concerning the description of methods, in Table 1, you refer to C* and H*, two indices that you have not defined at all in Section 3 (their formulas, their physical meaning), not to mention that you do not comment at all their changes between samples. The whole section in the Results part, concerning color attributes needs a serious re-consideration.

Another important weakness is the really bad quality of the figures, that need a much better analysis. Especially figures 1, 3 and 4 are almost unreadable.

Finally, in §2.3.1/2.3.2, you include measurements that are part of your previous work (e.g. biogenic amines), and I am not certain that one can easily follow the correlations presented without knowledge of the actual results.

Specific comments:

Line 14: avoid using the statement ‘for the first time’ since this is the follow up of your previous publication.

Line 95: you use often the word ‘biosafety’. Try to explain what you mean.

§2.1.1: since you refer to a reddish product (red wine), I assume that L* parameter is not the most representative color attribute. Refer also to general comments…no mention to results on C* and H*. Check and add comments. On the other hand, in lines 135-149, you comment on the change of overall color (ΔE*) without including these measurements in Table 1.

Line 178: What do you mean by ‘linear relationship”? How can you assess such a relationship (or any mathematical relationship) with only 2 measurements (time zero and 3 months storage)? Please, clarify.

Lines 180-191: What is the need to state your reasons for doing so? Usually, we follow the opposite path, from a general method to a more accurate and sensitive analytical procedure. I believe that this part is not necessary.

Lines 200-216: Based on these results, one can conclude that CP treated samples are more susceptible to loss of antioxidant activity at storage? Does that depend on CP processing conditions? Throughout the manuscript, there is no clear analysis of the effect of the processing conditions of Cold Plasma. It would be really interesting and essential to improve submission originality to study in depth the effect of the duration of the treatment, as well as that of the type of CP (this is a general concern for the current form of the submission that would necessitate a deeper statistical analysis).

Lines 465-491 (Conclusion part): Vague conclusions. What about the inhibition of microbial growth and the explicit role of CP processing conditions? Need to be more specific. Finally, what method would you suggest for the better preservation of the wine? In any case, to support any choice, it would be essential (maybe as a part of your future research) to perform a systematic shelf life study (probably following the Accelerated Shelf life testing principles), including sensory attributes (e.g. aroma and taste) that are crucial for wine consumers’ acceptance.

Round 2

Reviewer 1 Report

The manuscript of Niedźwiedź et al. wanted to study the influence of cold plasma to the quality and the shelf life of red wine. You made a lot of effort for improving the paper. Congratulation for this work!

Author Response

Thank you very much for such a detailed review of our work. We believe that the corrections made have provided it with much greater value. We would also like to thank you for appreciating the effort put into revising our work. 

Reviewer 3 Report

In the revised manuscript, authors incorporated in the text all of the suggestions and enriched to a significant extent the content of their work, by addressing major issues underlined. They performed a very thorough and detailed reconsideration of their manuscript, modifying -in some points- the structure and introduced new tables, offering a more readable information. The result is, in my opinion, a revised manuscript of improved quality. However, there are still some minor issues to be addressed, before proceeding to a final judgement for publication:

Line 78: correct in the parenthesis to (HHP) and pulsed electric fields (PEF)…an ‘s’ is written by mistake after the parenthesis

Keep a uniform way of writing color parameter ΔE* ( in Eq 1 where you define this term, it is written without an asterisk, ΔE, whereas both in Table 1 and throughout the text it is written with an asterisk).

Line 563: microorganisms

Line 565: do those numbers, namely 4.21 and 3.17 refer to log(N)? It is not clear, please explain

Author Response

Thank you very much for such a detailed review of our work. We believe that the corrections made have provided it with much more value. 

Reviewer:

In the revised manuscript, authors incorporated in the text all of the suggestions and enriched to a significant extent the content of their work, by addressing major issues underlined. They performed a very thorough and detailed reconsideration of their manuscript, modifying -in some points- the structure and introduced new tables, offering a more readable information. The result is, in my opinion, a revised manuscript of improved quality. However, there are still some minor issues to be addressed, before proceeding to a final judgement for publication:

Line 78: correct in the parenthesis to (HHP) and pulsed electric fields (PEF)…an ‘s’ is written by mistake after the parenthesis

Keep a uniform way of writing color parameter ΔE* ( in Eq 1 where you define this term, it is written without an asterisk, ΔE, whereas both in Table 1 and throughout the text it is written with an asterisk).

Line 563: microorganisms

Line 565: do those numbers, namely 4.21 and 3.17 refer to log(N)? It is not clear, please explain

Response: The minor errors noted have been corrected.